# An approach to identify gene-environment interactions and reveal new biological insight in complex traits

Xiaofeng Zhu [1] ✉, Yihe Yang [1], Noah Lorincz-Comi [1], Gen Li[1],
Amy R. Bentley [2], Paul S. de Vries [3], Michael Brown [3], Alanna C. Morrison [3],
Charles N. Rotimi [2], W. James Gauderman[4], Dabeeru C. Rao[5],
Hugues Aschard [6,7] & the CHARGE Gene-lifestyle Interactions Working Group*

There is a long-standing debate about the magnitude of the contribution of gene-environment interactions to phenotypic variations of complex traits owing to the low statistical power and few reported interactions to date. To address this issue, the Gene-Lifestyle Interactions Working Group within the Cohorts for Heart and Aging Research in Genetic Epidemiology Consortium has been spearheading efforts to investigate $G \times E$ in large and diverse samples through meta-analysis. Here, we present a powerful new approach to screen for interactions across the genome, an approach that shares substantial similarity to the Mendelian randomization framework. We identify and confirm 5 loci (6 independent signals) interacted with either cigarette smoking or alcohol consumption for serum lipids, and empirically demonstrate that interaction and mediation are the major contributors to genetic effect size heterogeneity across populations. The estimated lower bound of the interaction and environmentally mediated heritability is significant ($P < 0.02$) for low-density lipoprotein cholesterol and triglycerides in Cross-Population data. Our study improves the understanding of the genetic architecture and environmental contributions to complex traits.

Current genome-wide association studies (GWAS) focus on detecting genetic variants that lead to different phenotype means across genotype groups[1,2], and have identified a large number of genetic loci that, in some cases, explain large proportions of the trait's SNP-heritability[3–5]. While it is commonly agreed that complex traits are influenced by genetic and environmental factors and their interactions[6–9], there is a long-standing disagreement about the

magnitude of the $G \times E$ contribution to heritability because of different theoretical models and assumptions[10,11]. As pointed out in ref. 12, arbitrarily defined parameterizations of genetic effects with non-additive gene actions may explain the same degree of genetic variation as the currently prevailing additive model. Thus, while using additive genetic models such as polygenic risk scores to predict individual quantitative or qualitative phenotypes has become standard[5], these

[1]Department of Population and Quantitative Health Sciences, School of Medicine, Case Western Reserve University, Cleveland, OH, USA. [2]Center for Research on Genomics and Global Health, National Human Genome Research Institute, National Institutes of Health, Bethesda, MD, USA. [3]Human Genetics Center, Department of Epidemiology, School of Public Health, The University of Texas Health Science Center at Houston, Houston, TX, USA. [4]Division of Biostatistics, Department of Population and Public Health Sciences, University of Southern California, Los Angeles, CA, USA. [5]Center for Biostatistics and Data Science, Institute for Informatics, Data Science and Biostatistics, Washington University School of Medicine, St. Louis, MO, USA. [6]Institut Pasteur, Université Paris Cité, Department of Computational Biology, F-75015 Paris, France. [7]Program in Genetic Epidemiology and Statistical Genetics, Harvard T.H. Chan School of Public Health, Boston, MA 02115, USA.*A list of authors and their affiliations appears at the end of the paper. ✉ e-mail: xxz10@case.edu

models may not be fully informative in understanding genetic architecture.

Interactions are often studied secondarily in comparison to additive variance, whose advantage is to explain most of the correlations among relatives and fit natural selection model well[10,13]. Theoretical studies have demonstrated that a significant portion of variance can be explained by an additive model even when the genetic contribution to a phenotype is purely through $G \times E$[14]. This limitation is one of the key factors explaining the low power of approaches modeling interactions conditional on additive variance. As a result, studies focusing on detecting $G \times E$ at the genome-wide level are seldom considered as primary analyses. By contrast, the joint evidence of main genetic and $G \times E$ effects, in addition to the $G \times E$ alone, is tested in the Gene-Lifestyle Interactions (GLI) Working Group within the Cohorts for Heart and Aging Research in Genetic Epidemiology Consortium (CHARGE)[9,15], where only a modest number of genetic loci have been identified through testing for $G \times E$ alone[16–19]. The joint test limits our ability to determine to what degree the currently identified loci reflect evidence for $G \times E$ contribution, making it difficult to understand the precise interplay between genetic and environmental factors.

Concurrently, Mendelian randomization (MR) has been developed and widely applied to study causal relationships between risk exposures and outcomes in the post-GWAS era[20,21]. Although an MR approach has been used to explain $G \times E$[22], the underlying connection between testing pleiotropic variants in the MR framework and the detection of $G \times E$ is currently unclear. Here, we conceptually connect $G \times E$ with MR framework, illuminate their similarities and demonstrate that the test of horizontal pleiotropy in MR[23,24] can be used for detecting $G \times E$. Based on this principle, one can identify $G \times E$ using existing available GWAS and GWIS summary statistics. We applied this approach to the summary statistics from the Global Lipids Genetics Consortium study (GLGC, $n = 1.65$ M)[3] and the summary statistics in the interaction analysis with cigarette smoking and alcohol drinking in the CHARGE GLI working group[17], with replication using direct interaction tests performed in the UK Biobank (UKBB) data. Although the UKBB data accounted for about one third of sample in the GLGC consortium, theoretical work suggests that such replication is statistically independent (Supplementary Note)

## Results

### Testing $G \times E$ and mediation based on Mendelian randomization (MR)

Traditionally a genome-wide interaction study (GWIS) with an environmental exposure on a quantitative trait $Y$ is modeled through a linear regression:

$$Y = \beta_0 + \beta_1 G + \beta_2 E + \beta_3 G \times E + \epsilon, \quad (1)$$

where $\beta_1, \beta_2$ and $\beta_3$ correspond to the 'main' effect of $G$ (in the presence of $E$), the main effect of $E$ and the interaction effect of $G \times E$, respectively, and $\epsilon$ is a random noise. Here $G$, $E$, and $G \times E$ represent a genotype value, environmental factor, and their interaction respectively. For simplicity, we do not include any covariates, but it will not affect the general conclusion. The interaction effect is evaluated by the *direct test* statistic $T_{direct} = \hat{\beta}_3^2 / \text{var}\left(\hat{\beta}_3\right)$, where $\hat{\beta}_3$ refers the estimate from the regression model (1). Theoretical work indicates that the test statistics for the main effect $\beta_1 = 0$ and the interaction effect $\beta_3 = 0$ are correlated, with the correlation coefficient equal to $-\mu_E / \sqrt{\mu_E^2 + \sigma_E^2}$, where $\mu_E$ and $\sigma_E^2$ are the mean and variance of the environmental factor in the data[14]. However, the power of the direct test is usually low because of the collinearity between $G$ and $G \times E$ which induces a covariance between the estimates of $\beta_1$ and $\beta_3$. This covariance produces uncertainty (i.e. larger standard error) which by itself reduces power for testing either $\beta_1$ or $\beta_3$, even if the underlying true model includes $G \times E$ alone (i.e., $\beta_1 = 0$ and $\beta_3 \neq 0$)[10,14].

In practice, a GWAS is routinely conducted first when studying the genetic contribution to a trait, which is typically done through a linear regression model without including environmental factors, i.e.,

$$Y = \alpha_0 + \alpha G + \epsilon, \quad (2)$$

where we refer to $\alpha$ as the 'marginal' effect from a GWAS (in the absence of $E$) to differentiate from the main effect $\beta_1$ in model (1). We show that $\alpha - \beta_1 = \frac{\rho\sigma_{E1}}{\sigma_{G1}}\beta_2 + (\mu_{E1} + \frac{\rho\sigma_{E1}}{\sigma_{G1}})\beta_3$, where $\rho$ is the mediation contribution of $G$ through $E$, $\mu_{E1}$, $\sigma_{E1}$, and $\sigma_{G1}$ represent the environmental mean, standard deviation, and genotype standard deviation in GWAS data, respectively, suggesting that testing the hypothesis $H_0$: $\alpha - \beta_1 = 0$ for the difference between the marginal and main effects is equivalent to testing for the combined effect of $G \times E$ and mediation, and further reduces to testing for the $G \times E$ when $G$ and $E$ are independent (i.e., $\rho = 0$, Supplementary Note). This hypothesis can be tested by the statistic $T_{diff} = (\hat{\alpha} - \hat{\beta}_1)^2 / \text{var}(\hat{\alpha} - \hat{\beta}_1)$, where $\hat{\alpha}$, $\hat{\beta}_1$, and their corresponding standard errors are estimated from the GWAS and GWIS analyses, respectively. In fact, $T_{diff}$ and $T_{direct}$ are equivalent when GWAS and GWIS are performed in the same data. We verified this property using real data analysis in the GLI studies[17], from which the summary statistics of the marginal, main, and interaction effects are available and the marginal effect was obtained after adjusting for $E$. We observed that the correlation between the statistics of the $T_{diff}$ and the direct test is 0.98 for LDL and current smoking (Supplementary Fig. 5). However, GWAS is often performed in a much larger sample than the GWIS because of data availability. The environmental exposure may have different distributions in cohorts for conducting GWAS and GWIS (i.e., different mean and variance). Furthermore, models (1) and (2) are likely to be performed by two different groups of investigators, which will bring variation across studies in trait definitions, trait measurement procedures, quality control procedures, and covariates. Moreover, the summary statistics are obtained through meta-analyses in both GWAS and GWIS analyses, which can bring additional variation and confounding factors, including population stratification and cryptic relatedness, leading to a potentially invalid comparison between the marginal and main effects. In fact, it has been reported that the confounding of population stratification is not sufficiently corrected in large GWAS[25,26]. Therefore, directly using $T_{diff}$ to screen the genome can be biased even for testing the combined contribution of interaction and mediation.

To overcome this bias, we note that the marginal effect estimate $\hat{\alpha}$ and the main effect estimate $\hat{\beta}_1$ have a linear relationship,

$$\hat{\alpha} = \theta\hat{\beta}_1 + \frac{\rho\sigma_{E1}}{\sigma_{G1}}\hat{\beta}_2 + \left(\mu_{E1} + \frac{\rho\sigma_{E1}}{\sigma_{G1}}\right)\hat{\beta}_3, \quad (3)$$

where $\theta$ reflects the contribution of main effect to marginal effect, which converges to 1 when GWAS and GWIS are conducted using homogeneous measurements of phenotypes and environments ("Methods"). The genetic variants with no $G \times E$ and no mediation will fall on the regression line but the variants with $G \times E$ or mediation will depart from this line. We do not expect this pattern to be systematically impacted by the variation across studies. Therefore, we search the genetic variants that depart from this regression line to test the combined effect of $G \times E$ and mediation, providing $\theta$ can be correctly estimated. This idea is conceptually the same as the MR framework when we introduce a pseudo exposure $\widetilde{X}$, representing a polygenic score of the trait (Fig. 1). We do not need to construct this pseudo exposure in our analysis because we work directly on the summary statistics under the MR framework. We then estimate the causal effect $\theta$ of the pseudo exposure $\widetilde{X}$ on trait $Y$ in the MR framework and the

A. Mendelian Randomization Pleiotropy Test.

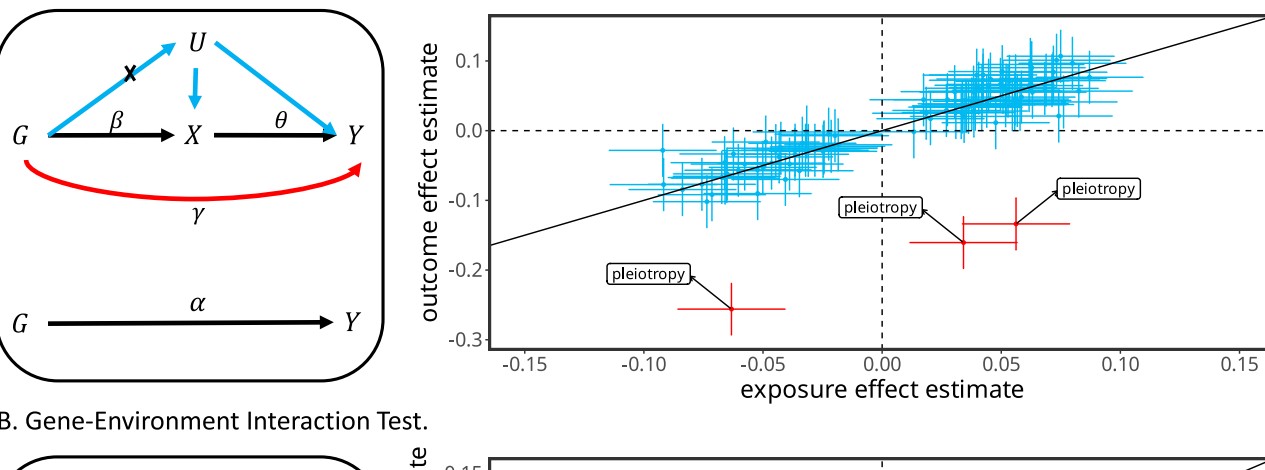

B. Gene-Environment Interaction Test.

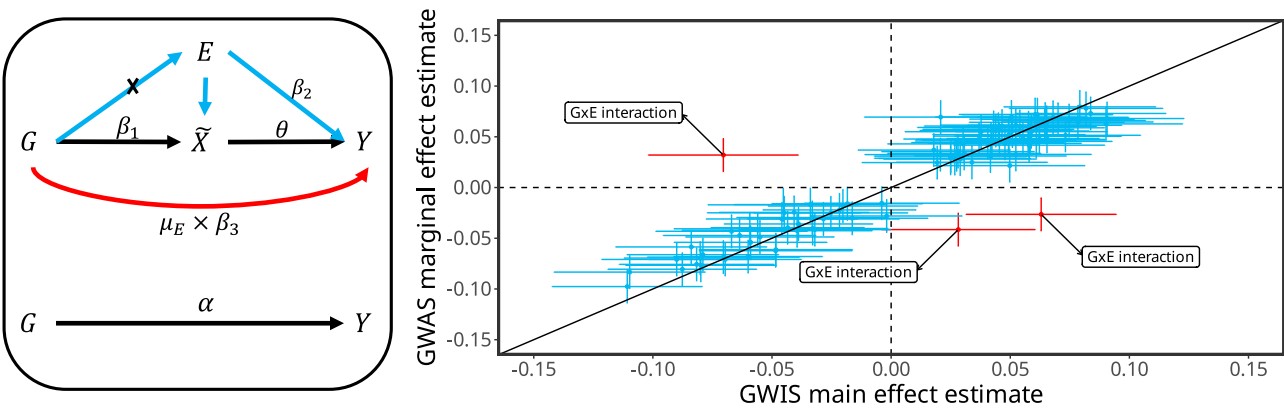

**Fig. 1 | Illumination of Mendelian randomization and $G \times E$. A** Left panel: the path diagram of the MR, where U refers to all confounders. Genetic variants ($G$) contributing to outcome $Y$ through mediation of exposure $X$ are often selected as the valid genetic instrumental variables (black paths). Genetic variants contributing to $Y$ through both black and red paths independently are horizontal pleiotropic variants. Genetic variants contributing to $Y$ through confounders (U) are invalid instrumental variables and need be blocked (x). Right panel: a scatter plot of effect sizes of genetic instrumental variants for an exposure and an outcome. Each + corresponds to the 95% confidence intervals of the exposure effect size (horizontal line segment) and the outcome effect size (vertical line segment). The horizontal pleiotropic variants (red +) depart from the regression line and can be separated from the variants with no pleiotropic effect (blue +). **B** Left panel: the $G \times E$ framework, with the goal of testing $G \times E$. Instead of an explicit exposure, we create a pseudo exposure $\tilde{X}$, which can be viewed as a polygenic score for trait $Y$ based on marginal effect sizes. However, our analysis does not require estimating this pseudo exposure. The genetic variants associated with the pseudo exposure $\tilde{X}$ but not through either the environment E or $G \times E$ are valid instrumental variables. The genetic variants interacting with E can be viewed the same as horizontally pleiotropic variants in the MR framework. Genetic variants associated with $Y$ via mediation through $E$ can contribute to both the pseudo exposure $\tilde{X}$ and $Y$, and thus have similar effects as $G \times E$ and cannot be distinguished from $G \times E$. Thus, testing the combined effect of interaction and mediation is conceptually equivalent with testing the horizontally pleiotropic effect in the MR framework. Right panel: a scatter plot of genetic variants for GWIS main effects and GWAS marginal effects. Each + corresponds to the 95% confidence intervals of the GWIS main effect size (horizontal line segment) and the GWAS marginal effect size (vertical line segment). Like the horizontal pleiotropic variants in the MR framework, $G \times E$ variants (red +) depart from the regression line and can be separated from variants with no $G \times E$ assuming no mediation.

$G \times E$ effect or mediation through $E$ is tested in the same way as testing for horizontally pleiotropic variants[23]. In doing so, we first select a set of independent variants associated with trait $Y$ and perform the inverse variance weighted analysis to estimate $\theta$, denoting as $\hat{\theta}$. Second, we test the $G \times E$ or mediation of a genetic variant through $E$ by the statistic $T_{MR\_GxE} = \frac{\left(\hat{\alpha} - \hat{\theta}\hat{\beta}_1\right)^2}{\text{var}\left(\hat{\alpha} - \hat{\theta}\hat{\beta}_1\right)} \sim \chi_1^2$. This test can be performed by the iterative Mendelian randomization and pleiotropy (IMRP) approach[23,27]. The statistic $T_{MRGxE}$ is an asymptotically unbiased test for testing the combined effect of $G \times E$ and mediation through $E$ (Supplementary Note).

**Two-step procedure for testing $G \times E$**

Note that $T_{MR\_GxE}$ likely tests for the combined effect of $G \times E$ and mediation unless $G$ and $E$ are independent (i.e., $\rho = 0$). To test for $G \times E$, we propose a two-step procedure by using $T_{MRGxE}$ to screen the whole genome and then performing $T_{direct}$ on the variants surviving the

$T_{MRGxE}$ screen. We set the significance level at $5 \times 10^{-8}$ for the first step ($T_{MR\_GxE}$ test), and the significance level at 0.05/X for the step 2 $T_{direct}$ test, where X is the number of independent significant variants in the first step/test. This two-step procedure can increase power at the screening step when there is interaction and mediation and increases power at the direct testing step by substantially reducing the multiple comparison burden. $T_{MRGxE}$ and $T_{direct}$ are not independent (Supplementary Note), and therefore, the variants detected by the two-step procedure could still reflect the contribution of mediation and $G \times E$, and it is necessary for further replication by performing $T_{direct}$ in independent data. To mitigate this problem, we can exclude the variants identified through GWAS of $E$, which could represent large mediation effect.

**Type I error rate and power of $T_{MR\_GxE}$ and the two-step procedure**

We first performed a series of simulations to investigate the type-I error rate and power of $T_{MRGxE}$ in the absence of mediation. In

simulations we observed that $E(\hat{\theta})$ is close to 1 and the estimate $\hat{\theta}$ converges to 1 when sample size increases, which is expected by theoretical prediction (Fig. 2A and Supplementary Fig. 6a). The direct estimate of the interaction effect $\beta_3$ as well as of $(\hat{\alpha} - \hat{\beta}_1\hat{\theta})/\mu_E$ is also unbiased (Fig. 2B, Supplementary Fig. 6b), although the standard error of $(\hat{\alpha} - \hat{\beta}_1\hat{\theta})/\mu_E$ is affected by the environmental means in GWAS and GWIS. When no mediation is present, the type-I error rates for both $T_{MR\_GxE}$ and the direct test are well controlled (Fig. 2C and Supplementary Fig. 6c)). The power of $T_{MRGxE}$ depends on multiple parameters, including $\mu_E$ and allele frequency in GWAS and GWIS and is less powerful than $T_{Direct}$ when the environmental mean in GWAS is lower (Fig. 2D and Supplementary Fig. 6d). Additional simulations for the estimates of $\hat{\theta}$, interaction effect $(\hat{\alpha} - \hat{\beta}_1\hat{\theta})/\mu_E$, type-I error rate and power are presented in Supplementary Figs. 7–9.

We next investigated the performance of $T_{Direct}$, $T_{MR\_GxE}$ and the two-step procedure when mediation is present and multiple variants are tested. We generated 20 independent variants with one variant having mediation, interaction, or both. All three tests have well controlled type I error rates when mediation is absent (Fig. 2E and Supplementary Fig. 10A). When mediation is present, the type-I error rate was still well controlled, although inflation can be observed for the two-step test and $T_{MRGxE}$ when $E$ contributes to 5% of the outcome variation and the samples between GWAS and GWIS are completely overlapped (Supplementary Fig. 10B, C). This inflation was caused by mediation and quickly disappeared when the overlapping rate between GWAS and GWIS subjects was reduced. The statistical power of $T_{MRGxE}$ and the two-step procedure for testing $G \times E$ was much more improved than $T_{Direct}$ when mediation was present (Fig. 2F and Supplementary Fig. 10D–F).

## Identifying gene-smoking and gene-alcohol drinking interactions to serum lipids

We applied the two-step procedure to search for genetic variants interacting with cigarette smoking and alcohol drinking for serum lipids, using the summary statistics of high-density lipoprotein cholesterol (HDL-C), low-density lipoprotein cholesterol (LDL-C), and triglycerides (TG) from the GLGC ($n = 1.65$ M) and the CHARGE GLI ($n = 134$ K). To mitigate the effects of mediation through cigarette smoking or alcohol drinking, we excluded all loci with $P$-value $< 5 \times 10^{-7}$ reported in the early GWAS of cigarette smoking status or alcohol drinking[28], which represent relatively large effect sizes of variants on cigarette smoking and alcohol drinking. We observed that $\hat{\theta}$ ranged from 0.92–1.33, 0.95–1.62, 0.83–1.25, 0.87–1.37, and 0.95–1.28 for European, African, Asian, Hispanic, and Cross-population data, respectively (Supplementary Data S1). The departure of $\hat{\theta}$ from 1 suggests that the phenotype treatments, analysis protocols, and corrections for population structure were not identical between the GLGC and CHARGE GLI consortiums. For example, CHARGE GLI performed a natural logarithmic transformation to the lipid measurements, whereas GLGC further performed an inverse normal transformation. The number of principal components (PCs) for correcting populations was also different between GLGC and CHARGE GLI. Despite these discrepancies, we did not observe an inflation for $T_{MRGxE}$, with the genomic control $\lambda$ values being close to 1 (range 0.93–1.05, Supplementary Data S2).

Using $T_{MRGxE}$ to screen the genome, we observed 15 genome-wide significant loci consisting of 17 independent signals ($P < 5 \times 10^{-8}$, $r^2 < 0.1$), including 4 and 5 loci for LDL-C, 7 and 5 loci for HDL-C, and 5 and 6 loci for TG, interacting with cigarette smoking and alcohol drinking or mediating through them, respectively (Fig. 3A–C, G–I, Supplementary Data S3a). All but 3 loci have been reported to be associated with either cigarette smoking or alcohol drinking in the recent largest GWAS study with over 3 million samples[29], suggesting the contribution of both $G \times E$ and mediation. Since we already excluded the cigarette smoking and alcohol drinking variants

identified from a relatively smaller study[28], these detected variants should represent modest mediation effects. Locus-specific plots of all significantly associated loci are presented in Supplementary Fig. 11, which suggest that multiple protein-coding genes are present in these loci. Strikingly, all the loci have previously been mapped to lipids traits except *RPL5P26* on chromosome 10. The $G \times E$ or mediation loci are clearly departing from most of the lipids-associated variants (Fig. 3D–F, J–L). The population-specific $T_{MRGxE}$ results are presented in Supplementary Fig. 12, which are also consistent with the Cross-population results, although the main contribution comes from the European population.

By applying the two-step procedure, we observed that 8 of the 17 independent signals are significant when using the direct test $T_{Direct}$ after correcting for the 17 tests and 4 environmental factors (Table 1, $P < 7.35 \times 10^{-4}$). In comparison to the direct test in GWIS, the two-step procedure identified more $G \times E$ signals for each of the three lipid traits and four environmental factors (Supplementary Data S3b). This provides additional support for the enhanced statistical power of the two-step procedure. The tissue enrichment analysis using the GWAS-based pathway analysis tools, MAGMA[30] and FUMA[31], suggest that these loci are enriched in liver, hippocampus, small intestine, and stomach tissues (Supplementary Fig. 13). Multiple loci were colocalized with expression quantitative trait loci (eQTLs) in the corresponding liver, lung, and blood tissues in the genotype-tissue expression database (GTEx)[32] (Supplementary Fig. 14).

## Independent replication

We next attempted to replicate the evidence for these 8 independent signals in the UKBB. Although the UKBB data accounted for about one third of samples in the GLGC consortium, the direct test statistic $T_{Direct}$ calculated in the UKBB is independent of $T_{MRGxE}$, so are the $T_{Direct}$ test statistics calculated in UKBB and CHARGE GLI, thus qualifying this as an independent replication (Supplementary Note). Six of the 8 signals are replicated in the UKBB after adjusting for 32 tests ($P < 1.56 \times 10^{-3}$), and 5 of them were genome-wide significant by the $T_{Direct}$ test in combined CHARGE GLI and UK Biobank data (Table 1). All 8 independent signals have the same interaction direction in CHARGE GLI and UKBB except *LPL*, which is not significant in UKBB (Supplementary Data S3a). The *CETP* and *SMARCA4* loci are the only two loci with no reported mediation evidence through either cigarette smoking or alcohol drinking.

We now aim to understand if the interaction evidence observed in this study has an alternative explanation[33] because of linkage disequilibrium (LD) with a variant which has causal effect on cigarette smoking or alcohol drinking. To examine this, we searched if there exists a variant(s) at each of the loci in Table 1 explaining the observed interaction evidence in the UKBB. However, we did not observe such variants (Supplementary Fig. 15), suggesting that the interaction evidence presented in Table 1 is genuine. In total, we identified 5 loci consisting of 6 independent signals that have evidence of interaction with either cigarette smoking or alcohol drinking.

## $G \times E$ interaction and mediation to SNP heritability

Since $\hat{\alpha} - \hat{\beta}_1\hat{\theta}$ refers to the combined interaction and mediation contribution to the marginal effect, we use $\hat{\alpha} - \hat{\beta}_1\hat{\theta}$ to estimate the heritability contributed by the interaction and mediation through the LD score (LDSC) regression[34]. Note that this heritability is a lower bound of the phenotype variance contributed by the $G \times E$ and mediation through E and is a part of the heritability estimated through the marginal effect, which is often referred to as the SNP-heritability in GWAS. In both Cross-Population (Fig. 4A) and European population (Fig. 4B), we observed significant interaction and mediation heritability ($P < 0.03$) with ever cigarette smoking for LDL-C, and alcohol consumption or cigarette smoking for TG, suggesting that the heritability estimates based on marginal effects also include significant

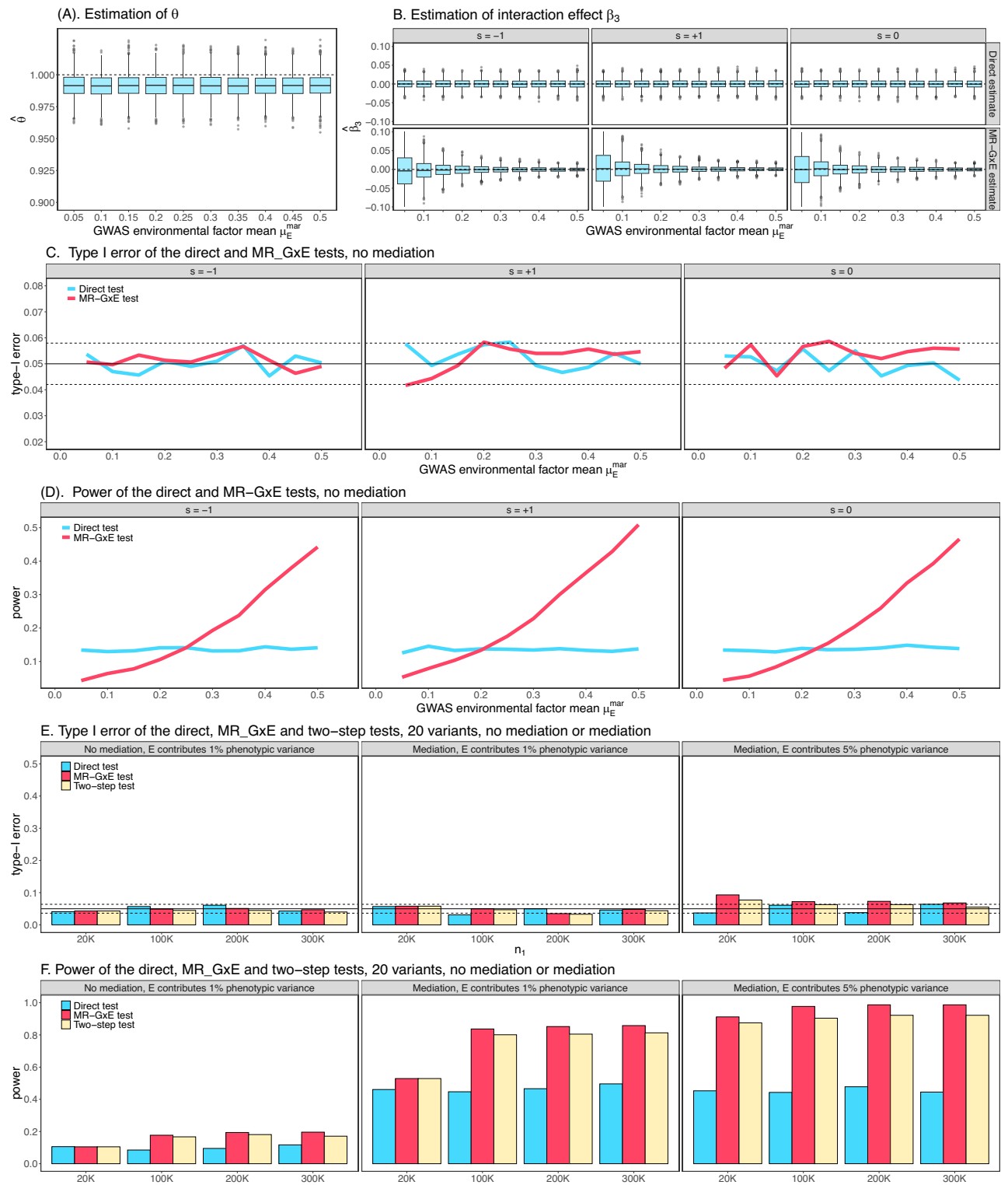

**Fig. 2 | Simulation performance of $T_{MR\_GxE}$ and the two-step procedure. A–D** No medication was present. The simulation details were described in "Methods". **A** Box plots of $\hat{\theta}$ in simulations under different environments in GWAs data. The top and bottom edges of the box plots represent the 25th and 75th percentiles of $\hat{\theta}$, and the horizontal middle line represents the 50th percentile. The vertical bars extend from the 25th (or 75th) percentile of $\hat{\theta}$ to the minimum (or maximum) value of simulated data. $E(\hat{\theta})$ is close to 1 as expected. **B** Box plots of the direct estimate of $\beta_3$ in GWIS (top panel) and by $(\hat{\alpha} - \hat{\beta}_1\hat{\theta})/\mu_e$ through MR-$G \times E$ analysis (bottom panel). The box plots are interpreted the same as in (**A**) accordingly. Both the estimates of $\beta_3$ and that by $(\hat{\alpha} - \hat{\beta}_1\hat{\theta})/\mu_e$ are unbiased. Here s = −1 refers to the scenario when the main effect and interaction effect have opposite effect directions; s = 0 refers to no main effect; and s = 1 refers to the scenario when the main effect and interaction effect have the same effect direction. **C** Type I error rate comparison between $T_{MR\_GxE}$ and the direct test for different main and interaction effect directions. Both $T_{MR\_GxE}$ and the direct test maintain the type I error rate well. **D** Power comparison between $T_{MR\_GxE}$ and the direct test for different main and interaction effect directions. **E**, **F** 20 variants were tested when mediation was present or not. The simulation details were described in "Methods". **E** Type I error comparison for $T_{Direct}$, $T_{MR\_GxE}$ and two-step procedure. The dash lines represent the 95% confidence interval. **F** Power comparison for $T_{Direct}$, $T_{MR\_GxE}$ and two-step procedure.

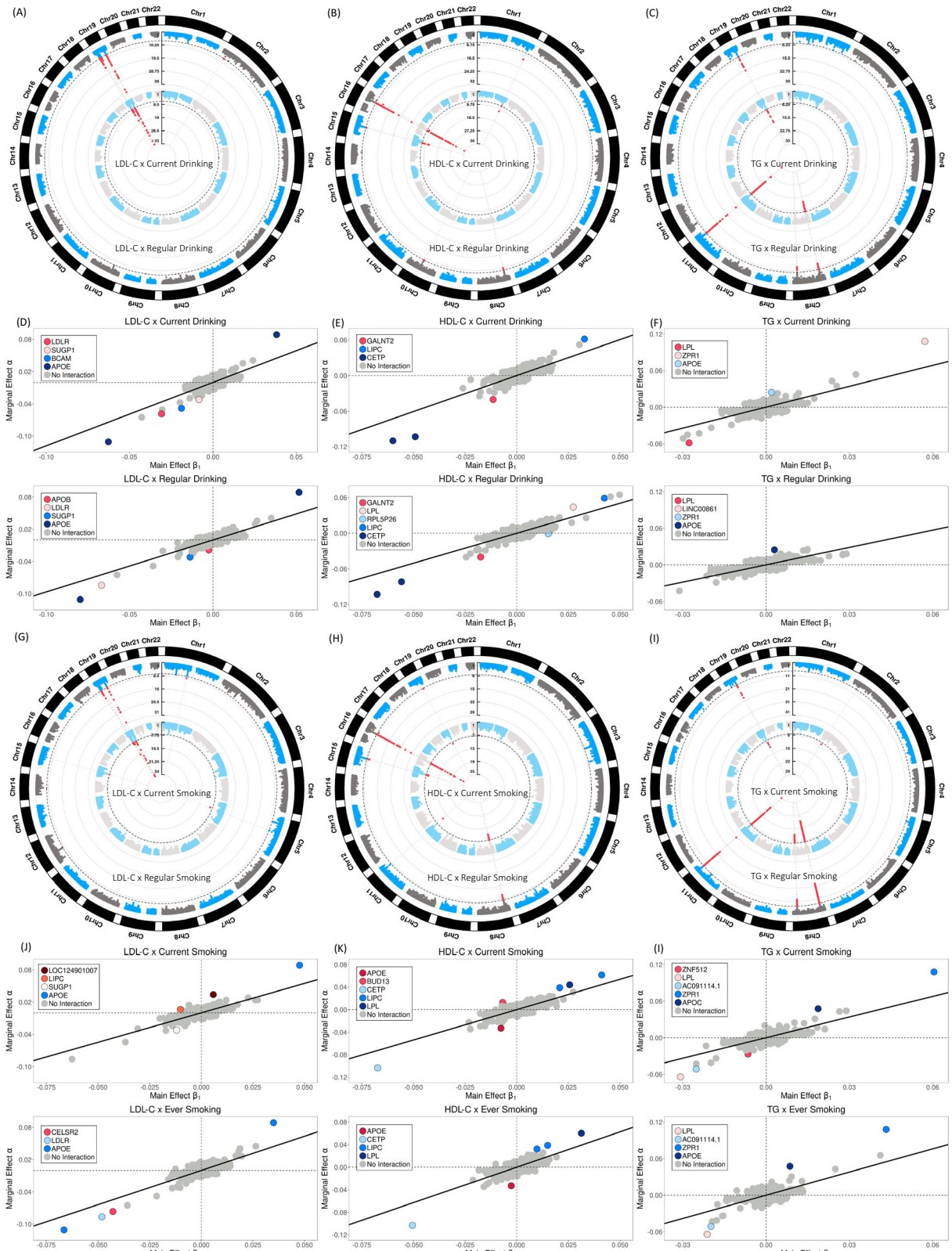

**Fig. 3 | Manhattan plots, marginal and main effect size comparisons.** The circle Manhattan plots of gene × alcohol drinking interactions for **A** LDL-C; **B** HDL-C; and **C** TG, respectively. The genome-wide significant loci are presented in red dots. The marginal and main effect sizes corresponding to alcohol drinking for **D** LDL-C; **E** HDL-C, and **F** TG, respectively. The colored circles represent the genome-wide significant loci and gray circles represent insignificant loci by $T_{MR\_GXE}$ test. The circle Manhattan plots of gene × cigarette smoking interactions for **G** LDL-C; **H** HDL-C; and **I** TG, respectively. The marginal and main effect sizes corresponding to cigarette smoking for **J** LDL-C; **K** HDL-C, and **L** TG, respectively.

**Table 1 | Interaction loci screened by $T_{MR\_GxE}$ and followed by the direct test $T_{Direct}$ in GLI (two-step test) and replicated by the direct test $T_{Direct}$ in UK Biobank**

| Mapping Gene | CHR: BP | Lead SNP | Environmental factor | Lipid traits | MR_GxE test P-value | GLI direct test P-value | UKBB direct test P-value | Combined GLI and UKBB direct test P-value |
|---|---|---|---|---|---|---|---|---|
| Signals identified by $T_{MR\_GxE}$ ($P < 5E{-}08$), by $T_{Direct}$ ($P < 7.35E{-}04$) and replicated by $T_{Direct}$ in UKBB ($P < 1.56E{-}3$) or combined GLI and UKBB $T_{Direct}$ $P < 5E{-}8$ | | | | | | | | |
| BUD13* | 11:116637146 | rs12294259 | Regular Drinking | TG | **2.47E−18** | **3.61E−06[a]** | **1.97E−04[a]** | **2.14E−08** |
| | 11:116657561 | rs3741298 | Current Smoking | TG | **2.80E−13** | **1.16E−10[a]** | 4.24E−01[a] | **6.99E−10** |
| CETP | 16:57000696 | rs8045855 | Current Drinking | HDL-C | **6.12E−24** | **1.85E−07[a]** | **4.97E−07[a]** | **4.05E−12** |
| | 16:57006829 | rs289713 | Regular Drinking | HDL-C | **5.01E−19** | **3.63E−07[a]** | **3.16E−06[a]** | **4.62E−11** |
| BCAM* | 19:45392254 | rs6857 | Regular Drinking | LDL-C | **4.02E−12** | **1.28E−06[a]** | **2.95E−04[a]** | **8.57E−09** |
| NECTIN2* TOMM40 APOE APCO1 | 19:45422946 | rs4420638 | Regular Drinking | LDL-C | **6.55E−36** | **4.41E−05[a]** | **1.95E−06[a]** | **2.08E−09** |
| LPL* | 8:19830170 | rs1569209 | Current Smoking | TG | **4.77E−10** | **1.01E−13[b]** | 3.49E−02[b] | **1.04E−13** |
| SMARCA4 | 19:11191677 | rs10402112 | Regular Drink | LDL-C | **1.85E−15** | **5.75E−04[a]** | **9.04E−04[a]** | 8.04E−06 |
| Signals identified by $T_{MR\_GxE}$ ($P < 5E{-}08$) and by $T_{Direct}$ ($P < 7.35E{-}04$) but failed in UKBB replication | | | | | | | | |
| RPL5P26* | 10:71533084 | rs11591480 | Regular Drinking | HDL-C | **3.34E−08** | **1.11E−04[a]** | 5.23E−02[a] | 8.69E−05 |
| ZPR1** | 11:116662579 | rs651821 | Ever Smoking | TG | **7.34E−17** | **3.44E−05[a]** | 6.87E−01[a] | 1.73E−04 |

The P-values of $T_{MR\_GxE}$ and $T_{Direct}$ are two-sided P-values based on Z-scores. The P-values in the last column (combined GLI and UKBB direct Test P-value) are calculated from a chi-square test with 4 degrees of freedom. All P-values were not adjusted for multiple comparisons. The bold P-values represent significant variants after adjusting for multiple comparisons.
*The locus has been reported to be associated with cigarette smoking.
**The locus has been reported to be associated with both cigarette smoking and alcohol drinking.
[a]The interaction effect direction is the same in GLI and UKBB. Detailed effect sizes and standard errors are presented in Supplementary Data S3a.
[b]The interaction effect direction is opposite in GLI and UKBB. Detailed effect sizes and standard errors are presented in Supplementary Data S3a.

contributions from $G \times E$ and mediation through the corresponding environment factors (Supplementary Data S4).

### $G \times E$ interaction and mediation to heterogeneity of genetic effect sizes across populations

As noted in Eq. (3), the marginal effect estimate of a genetic variant in GWAS consists of the $G \times E$ and mediation contribution when the $G \times E$ and mediation occur. Because of the environment heterogeneity across populations, we expected that the marginal effect sizes of the variants will be less correlated across populations for the variants with than without $G \times E$ interaction or mediation. We calculated the marginal effect size correlations between European, African, Hispanics, and Eastern Asian for these variants reported in Graham et al[3] after excluding the variants in Supplementary Data S3a where their $G \times E$ interactions or mediations were observed in this study. Similarly, we calculated the marginal effect size correlations for the variants in Supplementary Data S3a. We compared the correlation and observed a median of 24.4% drop of the cross-population correlation coefficient (Fig. 5), strongly support that $G \times E$ interactions or mediations contribute to the marginal effect size heterogeneity across populations.

## Discussion

In this study, we utilize marginal effects from GWAS to search for $G \times E$. We conceptually demonstrated the deep connection between detecting $G \times E$ and MR for causal inference. Although $T_{MRGxE}$ tests for the combined effect of $G \times E$ and mediation, the two-step procedure of $T_{MRGxE}$ followed by $T_{Direct}$ in fact tests for $G \times E$, and its statistical power is much improved because of the following reasons: (1) the step 1 $T_{MRGxE}$ can increase power when a genetic variant has a mediation effect through the environmental factor. In this case, we expect a larger difference between the marginal effect and the main effect (Eq. (3)) than no mediation. (2) The difference between the marginal and main effect can further increase when the environmental distributions between GWAS and GWIS cohorts are different (Eq. (3) and Fig. 1D). (3) At the two-step procedure, multiple comparison burden is significantly alleviated because only significant variants survived at step 1 need to be examined. As demonstrated in this study, the two-step procedure

identified 8 independent signals in comparing with two by the direct test in GWIS. This is also consistent with when comparing with the direct test in GWIS, the two-step procedure identified more $G \times E$ signals for each of the three lipid traits and four environmental factors (Supplementary Data S3b). Detecting $G \times E$ using direct tests can be biased by unmeasured confounders due to omitting covariates in the regression models[35], but the two-step procedure is robust because $T_{MR\_GxE}$ is not affected by confounders such as population structure. Considering the advantages of the two-step procedure, we view it as a complement rather than a replacement of the direct test. This perspective arises from the fact that the two-step test necessitates additional GWAS summary statistics and may be less powerful than the direct test in some situations (Fig. 1D).

Our study demonstrated that the current heritability estimates based on marginal effects could also include contributions from $G \times E$ and mediation through the corresponding environment factors (Fig. 4 and Supplementary Data S4). We excluded cigarette smoking- or alcohol drinking-associated variants identified from a large cigarette smoking and alcohol consumption GWAS of 1.2 million individuals[28] in our analysis, which mitigates the potential mediation contribution in the $T_{MRGxE}$ analysis. However, among the 15 loci identified by $T_{MRGxE}$, only three were not reported in the much larger recent cigarette smoking and alcohol consumption GWAS of 3.4 million individuals, suggesting mediation through cigarette smoking and/or alcohol consumption is still present but with modest effects. Among the six $G \times E$ variants identified, 4 are associated with either cigarette smoking or alcohol drinking, suggesting that the $G \times E$ variants are also likely to be mediated through E and the mediation improves power to detect $G \times E$. Furthermore, we demonstrated that the current SNP heritability estimates based on marginal effects could also include significant contributions from $G \times E$ and mediation through the corresponding environment factors for LDL-C and TG (Fig. 4 and Supplementary Data S4). We did not observe significant contributions from $G \times E$ and mediation to heritability for HDL-C, potentially attributable to the relatively small sample sizes in our GWIS. Since the LDSC regression[34] cannot be used to estimate $G \times E$ heritability, our estimates reflect the low bound of the interaction and environmentally mediated

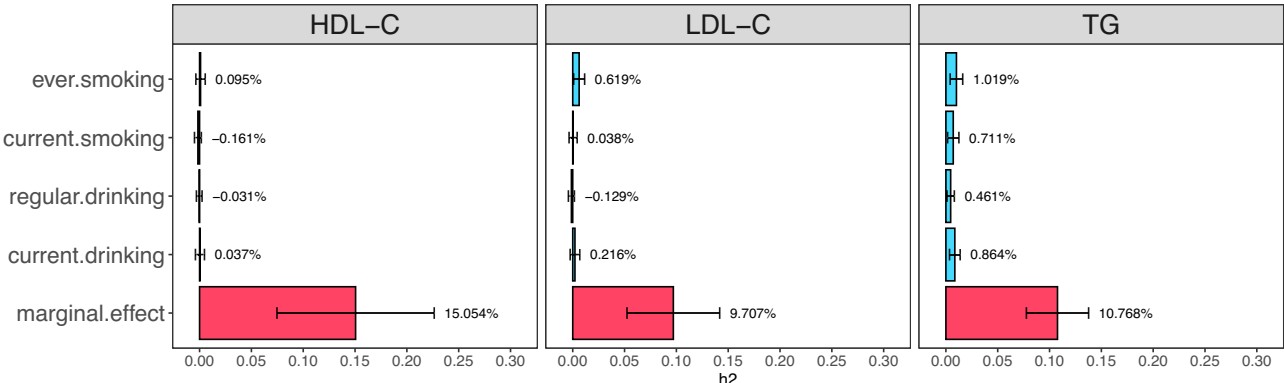

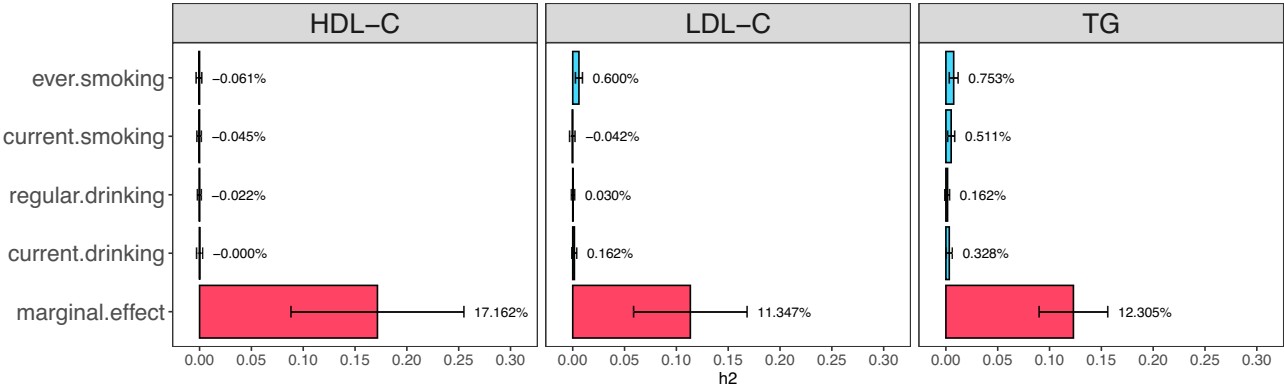

**Fig. 4 | The estimated heritability of HDL-C, LDL-C, and TG using LDSC regression. A** Cross-Population. **B** European population. X-axis represents heritability in percentage. Y-axis represents the corresponding heritability estimated in percentage (marginal.effect: marginal effect heritability; current.drinking: gene and current drinking interaction effect heritability; regular.drinking: gene and regular drinking interaction effect heritability; current.smoking: gene and current smoking interaction effect heritability; ever.smoking: gene and ever smoking interaction effect heritability). Marginal effect heritability refers to the heritability estimated through the marginal effect $\hat{\alpha}$, and interaction effect heritability refers to the heritability estimated through $\hat{\alpha} - \hat{\theta}\hat{\beta}_1$. The percentage number displayed on the right side of each bar represents the estimated heritability, and the corresponding 95% confidence interval shown as horizontal error bars. For the marginal effect analysis, the sample size is 1.65 M and 1.32 M for cross-population and European population analysis, respectively. For the interaction effect analysis, the sample size is 134 K and 80 K for cross-population and European population analysis, respectively.

heritability. We therefore suggest that the current SNP heritability estimates based on the marginal genetic effects be called marginal SNP heritability, to differentiate it from narrow-sense heritability[36] that is defined by additive genetic actions without the inclusion of $G \times E$ or mediation contributions. We believe this differentiation is important for correctly interpreting the current heritability estimates and understanding the genetic architecture of complex traits.

The 5 (6 independent signals) replicated loci interacting with cigarette smoking and alcohol consumption contain genes that are enriched in liver tissue, possibly reflecting the effect of alcohol drinking on aspartate amino transferase, alanine aminotransferase and γ-glutamyl transferase activities via the actions of numerous ingredients that alter the activities of enzymes found in the liver[37]. Among them, the interaction between alcohol consumption and cholesteryl ester transfer protein (*CETP*) has been reported for HDL-C and coronary artery disease[38–40]. The interaction between alcohol consumption and *APOE* on LDL-C has also been reported in a Mediterranean Spanish population[41], while the interactions between *APOA5* and cigarette smoking and alcohol drinking status associated with elevated TG and reduced HDL-C were observed in the Chinese and Korean populations[42,43]. However, our study is the only well-powered study demonstrating significant evidence at the genome-wide level and the interaction loci are replicable. *SMARCA4* was reported to be associated with LDL-C in the lipids GWAS in Africans[44]

but not in the recent largest lipids GWAS which is predominantly European ancestry[3]. Overall, the marginal effect sizes of the variants are less correlated across populations for the variants with than without $G \times E$ interaction or mediation (Fig. 5), empirically verified that $G \times E$ and mediation contribute to marginal effect differences across different populations[45]. We expect that including $G \times E$ interactions should improve polygenetic risk score prediction across populations.

It is well known that causal effect estimate in MR framework can be biased when the three IV assumptions are violated. However, our goal is to detect $G \times E$ rather than to estimate the causal effect. Detecting $G \times E$ based on MR is less likely to be biased for these reasons: (1) the effect sizes of IVs on the pseudo exposure are all highly significant in GWAS, which represent strong IVs. (2) It is less likely to have a confounding effect between a trait and its pseudo exposure, i.e., a polygenic score. (3) The iterative Mendelian randomization and pleiotropy test is a powerful method to detect pleiotropy when the two IV conditions are satisfied[23], in particular, it is expected that most of the IVs are not interacted with E. (4) Although the causal effect estimate can be affected if population structure is not well corrected, testing $G \times E$ by $T_{MRGxE}$ is not. The reason is that $T_{MRGxE}$ can be viewed as a weighted linear regression of the effect size of GWAS on the effect size of GWIS followed by searching the outlies of variants departing from the regression line. While the regression line (equivalent to causal

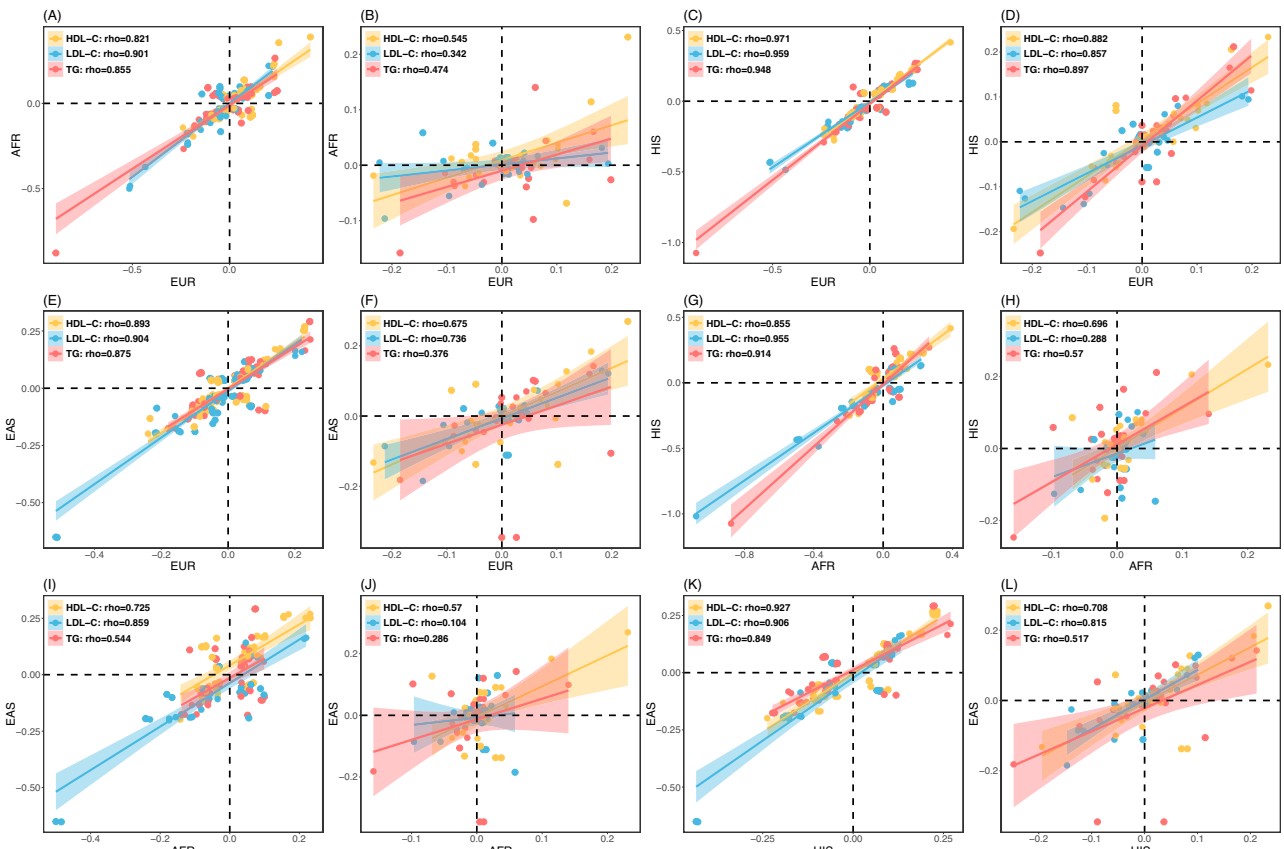

**Fig. 5 | Cross-population comparison of the LDL-C, HDL-C, and TG marginal effect sizes of the variants reported in Graham et al.[3].** **A** EUR vs AFR, no $G \times E$ interaction or mediation. **B** EUR vs AFR, $G \times E$ interaction or mediation. **C** EUR vs HIS, no $G \times E$ interaction or mediation. **D** EUR vs HIS, $G \times E$ interaction or mediation. **E** EUR vs EAS, no $G \times E$ interaction or mediation. **F** EUR vs EAS, $G \times E$ interaction or mediation. **G** AFR vs HIS, no $G \times E$ interaction or mediation. **H** AFR vs HIS, $G \times E$ interaction or mediation. **I** AFR vs EAS, no $G \times E$ interaction or mediation. **J** AFR vs EAS, $G \times E$ interaction or mediation. **K** HIS vs EAS, no $G \times E$ interaction or mediation. **L** HIS vs EAS, $G \times E$ interaction or mediation. The variants with no $G \times E$ interaction or mediation are those not included in Supplementary Data S3a. The variants with $G \times E$ interaction or mediation are those in Supplementary Data S3a. We only included independent variants. The shadow error bands represent the 95% confidence intervals. Clearly the variants without $G \times E$ interactions or mediations have substantially larger cross-population correlations than the variants with $G \times E$ interactions or mediations, suggesting that $G \times E$ interactions or mediations contribute the marginal effect size heterogeneity across populations. (European (EUR), African (AFR), Hispanics (HIS), Eastern Asian (EAS)).

effect estimate in MR) can be affected by population structure, the outlie detection is not.

In summary, our $G \times E$ approach is powerful and able to detect genetic loci interacting with environments that account for significant phenotypic variability. Our findings indicate that the contribution of $G \times E$ in lipids is not ignorable. Our study only focuses on the interactions of genes with cigarette smoking and alcohol consumption in lipids. The cumulative interaction contribution with many environmental factors can even be greater. Detecting individual genetic loci with environmental interactions facilitates a better understanding of the genetic architecture of complex traits and can improve phenotype prediction.

## Methods
### Summary statistics data
The marginal summary statistics of high-density lipoprotein cholesterol (HDL-C), low-density lipoprotein cholesterol (LDL-C), and triglycerides (TG) from the Global Lipids Genetics Consortium study (GLGC, $n = 1.65$ M)[3] were downloaded at http://csg.sph.umich.edu/willer/public/glgc-lipids2021.

GLGC consists of GWAS results from 1.65 M subjects representing five genetic ancestry groups: European ($N = 1.32$ M); African or admixed African ($N = 99$ K); East Asian ($N = 146$ K); Hispanic ($N = 48$ K); and South Asian ($N = 41$ K). We did not perform South Asian specific

analysis because there was no corresponding GWIS in the Cohorts for Heart and Aging Research in Genetic Epidemiology (CHARGE) consortium. The GWIS summary statistics from CHARGE gene-lifestyle (GLI) working group in this study are available via dbGaP (accession number phs000930). The CHARGE GWIS consists of 60 GWIS summary datasets: (LDL-C, HDL-C, and TG)-current smoking, (LDL-C, HDL-C, and TG)-ever smoking, (LDL-C, HDL-C, and TG)-current alcohol drinking, and (LDL-C, HDL-C, and TG)-regular alcohol drinking, for European, African or Admixed African, East Asian, Hispanic and multi-ancestry.

### QCs for performing $T_{MR\,GxE}$ analysis
To perform MR analysis, we aligned the GWAS summary statistics HDL-C, LDL-C, and TG from the GLGL with the corresponding GWIS summary statistics from the CHARGE gene-lifestyle consortium. We flipped the effect size if the corresponding reference allele did not match. We dropped a genetic variant if the two alleles were either {A, T} or {C, G}. We also excluded any variants with minor allele frequency (MAF) difference larger than 0.15 between GWAS and GWIS study. If multiple variants fall on the same chromosome position, we required the matched variants with MAF difference less than 0.01. We further excluded any variants with the effective sample size in GLGC trans-ethics or European less than 100 K and the other populations (African, Hispanic, East Asian) less than 30 K. To reduce the effect by mediations through

the smoking and alcohol drinking, we excluded all loci with $P$-value < 5E−7 identified by the GWAS of smoking status or alcohol drinking[28].

## $T_{MRGxE}$ analysis

To perform $T_{MRGxE}$, we applied the Mendelian randomization (MR) software IMRP[23] to estimate the causal effect by considering the main effect sizes from the GWIS of the CHARGE gene-lifestyle consortium as the exposure effects, and the marginal effects from the GLGC as the outcome effects, respectively. To identify instrument variables, we first selected all the variants with the $P$-value < 5E−8 after GC-correction in the GLGC, and then pruned them using the window size 500 KB and $r^2$ value 0.1 by the Plink software[46]. We standardized the effect sizes as in[27]. IMRP requires the input of the correlation coefficient to account for the effect of sample overlapping between GWAS and GWIS cohorts and this correlation was calculated based on the unsignificant variants ($P$-value > 0.05) across the genome. After estimate the causal effect, we performed $T_{MRGxE}$, which is equivalent to the pleiotropy test in the IMRP, to all the genetic variants across the genome.

## Independent locus definition

Independent loci were defined as the regions within 1 Mb of the most significant variants by the $T_{MRGxE}$ test. Independent signals were defined as the variants in a locus with $r^2$ < 0.1. The 1000 G data was used as the reference genetic data for LD calculation.

## Choosing independent variants for replication in UK Biobank

By applying $T_{MRGxE}$, we observed that 15 genome-wide significant loci consisting of 17 independent signals ($P$-value < 5E−8), including 4 and 5 loci for LDL-C, 7 and 5 loci for HDL-C, and 5 and 6 loci for TG, interacted with alcohol drinking and cigarette smoking, respectively (Supplementary Data S3a). At a locus with the $T_{MRGxE}$ significant ($P$-value < 5E −8) for a lipid trait (LDL-C, HDL-C, or TG) and environment (smoking or alcohol drinking), we searched the variant with the smallest $P$-value of the direct test $T_{Direct}$ among the significant variants by the $T_{MRGxE}$. The variants with $T_{Direct}$ $P$-value < 7.35E−4, which correct for the 17 tests and 4 environmental factors, were considered as significant for $G \times E$ interaction (two-step procedure). We obtained 8 independent variants in 6 loci among these 17 independent signals survived the threshold P-value = 7.35E−4 and these variants were further tested for the replication of the interaction effects in UK Biobank using $T_{Direct}$ test.

## LD score regression

We applied the LD score regression to estimate heritability contributed by $G \times E$ interaction and mediation through the environment factor $E$. We estimated heritability by combining all chromosomes rather than chromosome specifically. We used the R package bigsnpr[47] to estimate LD scores in the corresponding populations from 1000 G reference data with default settings.

## Functional mapping and annotation

We performed overall enrichment tests using the residual $\hat{\alpha}_j - \hat{\beta}_j\hat{\theta}$ as the effect size and $se(\hat{\alpha}_j - \hat{\beta}_j\hat{\theta})$ as the corresponding standard error. We used MAGMA[30] (Multi-marker Analysis of GenoMic Annotation) and DEPICT[48] (Data-driven Expression Prioritized Integration for Complex Traits) to identify tissues and cells that are highly expressed at genes within the $G \times E$ loci. We also used DEPICT to test for enrichment in gene sets associated with gene ontology (GO) ontologies, mouse knockout phenotypes and protein-protein interaction networks. In addition, we reported significant enrichments with a false discovery rate 0.05. Analysis was done using the online platform FUMA GWAS.

## Colocalization

We performed colocalization analysis by using the software ezQTL (https://analysistools.cancer.gov/ezqtl/#/home). We chose the

public genotype-tissue expression (GTEx) v7 with eQTL[32] as the QTL data and chose the public European reference panels for calculating the LD data. We performed colocalization analysis between GWIS and QTL results within a locus using eCAVIAR (eQTL and GWAS Causal Variant Identification in Associated Regions)[49], where the Colocalization Posterior Probability (CLPP) is used to describe the significance level of colocalization. We only recorded colocalization with CLPP > 0.01, as suggested by the authors of eCAVIAR.

## UK Biobank individual level data for replication

The UK Biobank (UKBB)[50] individual-level data used for replications were available through Application ID: 81097. Quality Controls Participants in the UKBB were genotyped using a custom Affymetrix UK Biobank Axiom array[51]. Genotypes were imputed by the UKBB using the Haplotype Reference Consortium reference panel[52] with imputation $r^2$ value greater than 0.3. Related individuals with pairwise kinship coefficient greater than 0.0884 (suggested by UKBB) were removed from analysis, resulting in $N = 445,424$ individuals of European, African, and East Asian ancestries. The principal components were calculated by UKBB with genotype data within each ancestry to account for population structure. These data were independent of GLI cohorts and consisted of European, African, and Asian individuals (race determined using UKBB field ID 21000-0.0) in UKBB who were unrelated (genetic kinship coefficient less than 0.0884; 22021-0.0). Linear regression model (1) in main text was performed. Covariates included age at assessment (21003-0.0), age², sex (31-0.0), the first 10 PCs (22009-0.1 to 22009-0.10), and environment exposure, a genetic variant and their interaction. Environmental exposures included ever/never smoking status (20116-0.0), current/non-current smoking status (20116-0.0), and alcohol intake frequency (1558-0.0).

Analogous to the $G \times E$ analysis in ref. 17, HDL-C (30760-0.0) and TG (30870-0.0) measurements were natural log transformed and LDL-C measurements (30780-0.0) were converted from mmol/L to mg/dl then multiplied by a factor of 0.7 if there was a history of lipid-lowering medication (6177-0.0) present. LDL-C measurements were therefore considered no medication if there were missing values for medication history. This introduced missing values in LDL-C for 248,419 individuals.

## Theoretical properties of $T_{MRGxE}$

In MR analysis, the instrumental variables are independent and are genome-wide significant variants selected from GWAS. Let $\hat{\beta}_{1j}, \hat{\beta}_{2j}, \hat{\beta}_{3j}$ and $\hat{\alpha}_j, j = 1, \ldots, m$, be the corresponding effect size estimates in GWIS (model (1) and GWAS (model 2)) for the m instrument variables.

The causal effect $\theta$ of the inverse variance weighted (IVW) is estimated by

$$\hat{\theta} = \arg\min_{\theta} \left\{ \frac{1}{m} \sum_{j=1}^{m} \frac{\left( \hat{\alpha}_j - \hat{\beta}_{1j}\theta \right)^2}{\text{var}\left( \hat{\alpha}_j \right)} \right\}. \tag{4}$$

It is much simply to work on $\hat{\theta}$ by standardizing the IVs and this procedure does not change the conclusion. Thus, we let $\sigma_{G,j}^2 = 1, j = 1, \ldots, m$, in both GWAS and GWIS data. Further, we let the phenotype residue variance $\sigma^2 = 1$. By equation (S15) in Supplementary Note, we

have $\text{var}(\hat{\alpha}_j) = n_1^{-1} j = 1, \ldots m$, and $\hat{\theta} = \frac{\sum_{j=1}^{m} \hat{\alpha}_j \hat{\beta}_{1j}}{\sum_{j=1}^{m} \hat{\beta}_{1j}^2}$.

Since only the variants without either $G \times E$ interaction or mediation are valid in the MR analysis, we assume $\rho = 0$ (no mediation) and

$\beta_{3,j} = 0$ (no interaction). We have

$$\hat{\alpha}_j = \hat{\beta}_{1,j} + \mu_{E1}\hat{\beta}_{3,j} \qquad (5)$$

By applying the Slutsky's theorem, and let $\beta_{3,j} = 0$, we have:

$$E\left(\hat{\theta}\right) = \frac{\frac{1}{m}\sum_{j=1}^{m} E(\hat{\alpha}_j\hat{\beta}_{1,j})}{\frac{1}{m}\sum_{j=1}^{m} E(\hat{\beta}_{1,j}^2)} = \frac{\frac{1}{m}\sum_{j=1}^{m} \beta_{1,j}^2 + \frac{n_0}{n_1 n_2}\left(1+\frac{\mu_{E0}^2}{\sigma_{E0}^2}\right)}{\frac{1}{m}\sum_{j=1}^{m} \beta_{1,j}^2 + \frac{1}{n_2}\left(1+\frac{\mu_{E2}^2}{\sigma_{E2}^2}\right)}. \qquad (6)$$

Because $\sigma_{G,j}^2 = 1$, $\frac{1}{m}\sum_{j=1}^{m} \beta_{1,j}^2$ is the average phenotypic variance accounted by an IV. Define $\sigma_\beta^2 = \frac{1}{m}\sum_{j=1}^{m} \beta_{1,j}^2$, we have:

$$E\left(\hat{\theta}\right) = \frac{\sigma_\beta^2 + \frac{n_0}{n_1 n_2}\left(1+\frac{\mu_{E0}^2}{\sigma_{E0}^2}\right)}{\sigma_\beta^2 + \frac{1}{n_2}\left(1+\frac{\mu_{E2}^2}{\sigma_{E2}^2}\right)}, \qquad (7)$$

which converges to 1 when $n_1$ and $n_2 \to \infty$. However, when $\sigma_\beta^2$ is small (weak instrument in MR analysis), the converge of $E(\hat{\theta})$ to 1 is slow. We also note that $E(\hat{\theta}) \le 1$.

### Simulation settings without medication contribution (Fig. 2A–D, Supplementary Figs. 6–9)

For $i$th individual, we generated $m = 102$ number of independent variants through for $j = 1, \ldots, m$ by $G_{ij}^* \sim \text{Binom}(2, p_j)$, where $p_j \sim \text{unifom}(0.05, 0.5)$. We standardized genotypes by $G_{ij} = \frac{G_{ij}^*}{2p_j(1-p_j)}$. For the environment factor in the GWAS model, we generated $E_{i1} \sim \mathcal{N}(\mu_{E1}, 1)$. For the environment factor in the GWIS model, we generated $E_{i2} \sim \mathcal{N}(\mu_{E2}, 1)$. For the samples overlapped between the GWAS and GWIS, we generated their environment values through $\mathcal{N}(\mu_{E2}, 1)$. We varied the values of $\mu_{E1}$, $\mu_{E2}$ and the proportion of overlapped samples.

The main effect size of the $j$th variant was generate by $\beta_{1j} \sim \mathcal{N}(0, \sigma_\beta^2)$, where $\sigma_\beta^2$ is the trait variance accounted for by the IVs. For the first variant, we added its interaction effect with E. The phenotype $Y_i$ by generated by

$$Y_i = \sum_{j=1}^{m} G_{ij}\beta_{1j} + 0.1E_i + 0.05(G_{i1} \times E_{i1}) + \epsilon_i, \qquad (8)$$

where $\epsilon_i \sim \mathcal{N}(0, \sigma^2)$. The causal effect $\theta$ was estimated using the last 100 variants as the IVs. The power and type I error rate for $T_{Direct}$ and $T_{MR\_GxE}$ were calculated based on the first and second variants, respectively.

### Simulation settings without medication contribution (Fig. 2E–F), Supplementary Fig. 10

We generated 20 independent variants by $G_j \text{ Binom}(2, 0.3)$ and standardized it but without mean correction. We simulated environment E according to mediation present or not present. If no mediation, $E$ is generated from $\mathcal{N}(1,1)$. If there is mediation, $E \sim 0.05G + \mathcal{N}(2, 0.9975)$, or $G$ contributes 0.25% variation of E. The phenotype is generated according to the following models:

1. No mediation and no interaction: $Y \sim 0.1G + \gamma E + N(0,10)$, where $E \sim \mathcal{N}(1,1)$
2. Mediation but no interaction: $Y \sim 0.1G + \gamma E + N(0,10)$, where $E \sim 0.05G + \mathcal{N}(1, 0.9975)$.
3. Mediation and interaction: $Y \sim 0.1G + \gamma E + 0.1*G*E + N(0,10)$, where $E \sim 0.05G + \mathcal{N}(1, 0.9975)$.

We let $\gamma$ take values of 1 and $\sqrt{5}$. We also simulated data with environment mean 0.5 (Supplementary Fig. 10). We first simulated $n_2 = 20,000$ subjects for GWIS cohort (or main effect estimation). The sample size for marginal effect estimation varied from $n_1 = 20,000$ to 300,000, with the 20,000 subjects in GWIS cohort was always included. For the non-overlapped subjects, we let the environment mean to be 1.5 times of the environwide mean in GWIS cohort. The type I error and power for $T_{Direct}$ and $T_{MR\_GxE}$ were calculate by correcting for 20 tests using the Bonferroni correction. For the two-step procedure, we first applied $T_{MR\_GxE}$ and Bonferroni correction. The variants survived after $T_{MR\_GxE}$ were further tested by $T_{Direct}$ and Bonferroni correction was further applied.

### Reporting summary

Further information on research design is available in the Nature Portfolio Reporting Summary linked to this article.

### Data availability

The marginal summary statistics of HDL-C, LDL-C, and TG from the Global Lipids Genetics Consortium study (GLGC, $n = 1.65$ M) were downloaded at http://csg.sph.umich.edu/willer/public/glgc-lipids2021. GLGC consists of GWAS results from 1.65 M subjects representing five genetic ancestry groups: European ($N = 1.32$ M); African or admixed African ($N = 99$k); East Asian ($N = 146$k); Hispanic ($N = 48$k); and South Asian ($N = 41$k). We did not perform South Asian specific analysis because there was no corresponding GWIS in the Cohorts for Heart and Aging Research in Genetic Epidemiology (CHARGE) consortium. The GWIS summary statistics from CHARGE gene-lifestyle (GLI) working group in this study are available via dbGaP (accession number phs000930). The UKBB individual-level data for replications were available through Application ID: 81097.

### Code availability

$T_{MRGxE}$ test was used the software IMRP which is available in the Github repository with the following link, https://github.com/XiaofengZhuCase/IMRP[23]. Heritability analysis was performed by Bigsnpr, https://privefl.github.io/bigsnpr/reference/snp_ld_scores.html[47] and LDSC regression, https://github.com/bulik/ldsc[47]. FUMA: https://fuma.ctglab.nl/[31]. Software ezQTL: https://analysistools.cancer.gov/ezqtl/#/home. MAGMA: https://ctg.cncr.nl/software/magma[30]. DEPICT: https://github.com/perslab/depict[48]. Plink: https://www.cog-genomics.org/plink2/[46]. The codes for performing simulations and analyzing $G \times E$ interaction of lipids data were deposited in the Zenodo database[53].

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

## Acknowledgements
This work was supported by grant R01HG011052 and R01HG01152-03S1 (to X.Z.) from the National Human Genome Research Institute (NHGRI) and R01HL118305 and R01HL156991 (to D.R.) from the National Heart, Lung and Blood Institute. This work was also supported in part by the Intramural Research Program of NHGRI through the Center for Research on Genomics and Global Health (CRGGH). The CRGGH is also supported by the National Institute of Diabetes and Digestive and Kidney Diseases and the Office of the Director of the National Institutes of Health (Z01HG200362 to A.R.B.).

## Author contributions
X.Z. conceived and designed the study. X.Z., Y.Y., N.L., and G.L. performed analysis. X.Z. drafted the initial manuscript with inputs from others. X.Z., Y.Y., N.L., G.L., A.R.B., P.S.d.V., M.B., A.C.M, C.N.R., W.J.G., D.R., and H.A. critically revised and approved the manuscript.

## Competing interests
The authors declare no competing interests.

## Additional information

## the CHARGE Gene-lifestyle Interactions Working Group

Xiaofeng Zhu [1] ✉, Amy R. Bentley [2], Paul S. de Vries [3], Michael Brown [3], Alanna C. Morrison [3], Charles N. Rotimi [2], W. James Gauderman[4], Dabeeru C. Rao[5] & Hugues Aschard [6,7]

