## [Peer Review File · Nature Communications]

An Approach to Identify Gene-Environment Interactions and Reveal New Biological Insight in Complex traitsREVIEWER COMMENTS

Reviewer #1 (Remarks to the Author):

The authors proposed a new and powerful method to test for G-E interactions, a notoriously difficult problem. The main idea is EXREMELY novel: they reformulated the original problem of testing for G-E as testing for (horizontal) pleiotropy in Mendelian randomization (MR). Simulation and real data studies clearly demonstrated the power gains and thus potential wide applications of the proposed method over the existing "direct method" of directly testing the interaction in a joint model.

I only have a few MINOR comments or clarification questions.

1. "a pseudo exposure" in the text and Figure 1 but also referred to as "a polygenic score" e.g. in Line 358. Should it be the same outcome Y but based on a GWIS? If so, perhaps better to refer it so and consistently in both the text and Fig 1; otherwise, how is the polygenic score constructed? And does it matter which PRS is used?

2. What is/are the nominal significance level/levels being used in MR_GxE and each of the two tests/steps in the two-step procedure? Does it matter whether/how to adjust for multiple testing?

3. In the Discussion, it is stated that " $T_{\{MR_GxE\}}$ is not affected by confounders such as population structure" and "It is less likely to have a confounding effect between a trait and its pseudo exposure": Wouldn't population structure (PS) influence most of MR? In particular, why wouldn't PS influence IMPR?

4. Is there any simple or intuitive explanation on why the proposed method is more powerful than the direct testing method? This is relevant because causal inference through MR seems to be a more difficult problem. Some comments would be helpful.

Reviewer #2 (Remarks to the Author):

Zhu and colleagues present a new method for detecting GxE interactions (they do not appear to have a name for this new method, so I will simply refer to it as NewTest). The authors show that NewTest is well-calibrated (controls Type 1 error), and that it has the potential to be more powerful than the conventional test for GxE interactions (called GWIS). Specifically, NewTest has two potential advantages over GWIS. Firstly, NewTest uses a two-step approach, that first finds loci with significant non-additive effect, then considers only these loci when considering interaction with the environment (which reduces the multiple testing burden). Secondly, NewTest can combine GWIS results with those from marginal analysis (my instinct is that this enables more accurate estimation of the marginal effect, that in turn increases power of the interaction test).

It seems that (at present) the utility of NewTest is relatively slight. Specifically, the results presented indicate that, for the three lipid traits considered, the benefit of NewTest over GWIS is relatively small, and perhaps even negative (i.e., NewTest finds 10 significant loci, whereas GWIS found 15 or 17). Furthermore, to use NewTest, it is necessary to first perform a GWIS (so NewTest is not a standalone replacement for GWIS).

Despite these criticisms, I am generally supportive of the paper. This is because I think the method is novel, and I am very interested by the connection between GWIS and MR.

As an aside, while I consider the paper well-written, I nonetheless found it complicated, and so apologise if I have made a major error in my understanding of your work.

###

Major Comments

1 - I would appreciate a more explicit comparison of the powers of NewTest and GWIS on real data. For example, you could have a table reporting the number of SNPs significant from NewTest and GWIS (and the overlap) for each of the three traits.

2 - I explain above what I consider the two advantages of NewTest over GWIS. If my understanding is correct, can you quantify the relative importance of each. My opinion is that the first (reduction of multiple testing) is much larger than the second (ability to incorporate marginal results from larger GWAS). If so, this might be "good news", in the sense that I imagine there are relative few phenotypes where we have distinct results from GWIS and GWAS. For the real data results, it might be interesting to run NewTest using only the GWIS results (to see whether including GWAS results had much impact).

3 - It seems to me, that for the real phenotypes considered, NewTest is less powerful than GWIS. If correct, this creates a contradiction with your simulations, where NewTest is almost always more powerful than GWIS (to use your notation, "Two-Step test" is more powerful than "Direct test"). This suggests your simulations are unrealistic (at least wrt to lipid traits). Given Fig 2D, it might suggest that the environmental factor tends to be very small for real traits.

4 - I would appreciate if you made clear what you think the use of NewTest is. In my opinion, it is hard to view NewTest as a replacement for GWIS (because it gives, at most, a small increase in power, and requires a GWIS to be run first). If you agree, then I think good to make this explicit.

5 - Regarding the heritability estimates. Firstly, I am not aware of a paper that explicitly shows that it is valid to apply LDSC to GWIS summary statistics (and that the resulting estimate is an estimate of GxE heritability). The basic use of LDSC is to estimate h^2_{SNP} , and works because if SNP j contributes one unit of heritability, then SNP k will tag $\text{cor}(X_j, X_k)^2$ of this. Does the same principle hold when considering interactions? (i.e., if SNP j contributes one unit of GxE heritability, then SNP k will tag $\text{cor}(X_j, X_k)^2$ of this?). Secondly, if it is valid to use LDSC to estimate GxE heritability, is there an advantage of using summary statistics from NewTest instead of those from GWIS (I can believe there is - perhaps using the former is better, because NewTest is less restrictive - but this is not clear from the text). If you retain the heritability analyses, I think it would be good to compare estimates using NewTest and GWIS summary statistics.

6 - Also regarding the heritability analysis. On Line 539, you say " We observed that some of the chromosome specific heritability estimates were negative. We only summed the non-negative chromosome specific heritability estimates." this is not satisfactory. Firstly, it sounds like you performed 22 separate analyses (one for each chromosome), then combined these. If so, it is not clear why (you say "to account for potential heterogeneity"). While there is a potential advantage in dividing by chromosome (e.g., if one chromosome has a much larger effect than others), I believe this should be done within a joint analysis (ie run LDSC once using 22 partitions) and not as 22 separate analyses. Further, it is generally bad practice to ignore negative estimates, as this leads to upward biases (instead, if you perform a joint analysis, you will get the per-chromosome estimates plus their sum, and a corresponding SD).

Minor Comments

7 - L 102 - "Instead, the joint evidence of main genetic and GxE effects, in addition to the GxE alone," - Do you mean, "by contrast" (you have just explained that few studies consider interactions, or at least not as the primary goal ... I think you are then saying GLI is one of the exceptions, in that it

does focus on interaction)

8 - L 115 - "Based on this principle, one can identify novel GxE using existing summary statistics without needing costly and time-consuming new analyses from all cohorts." I feel this downplays the resources needed - it remains that NewTest requires summary statistics from a GWIS (which as the authors have just explained, are not widely available).

9 - L152 - not sure you have defined rho

10 - L178 - "this pattern will not be impacted by the systemic variation across studies". It seems certain that experimental differences / noise, will have an impact, but I think your point is that it is unlikely to introduce systematic bias. So maybe this sentence should be "... we do not expect this pattern to be systematically impacted ..."

11 - L 194 - "applying ... to screening " - maybe "better" english is "by using ... to screen "

12 - L 319 - "Our study demonstrated that the current heritability estimates based on marginal effects also include significant contributions from \times and mediation through the corresponding environment factors (Fig. 4 and Table S4)". It seems that you first discuss the heritability results in the Discussion (whereas I feel they should be in the results)

13 - L 443 - "No medication was present" (should be "mediation")

14 - L 495 - Please describe the change summary statistics (i.e., that there are six sets, corresponding to GWIS of tg, hdl and ldl versus smoking or alcohol)

15 - L 500 - "we merged the" - perhaps better to say "aligned" (because else it sounds like you did a meta analysis of the GWAS and GWIS results)

16 - I really appreciated how much effort you put into the supplementary material.

###

Signed Doug Speed

We appreciate the reviewers' outstanding comments. We have revised our manuscript by providing point by point response to the comments. The changes in the revised version of the manuscript are highlighted in red.

Reviewer #1 (Remarks to the Author):

The authors proposed a new and powerful method to test for G-E interactions, a notoriously difficult problem. The main idea is EXTREMELY novel: they reformulated the original problem of testing for G-E as testing for (horizontal) pleiotropy in Mendelian randomization (MR). Simulation and real data studies clearly demonstrated the power gains and thus potential wide applications of the proposed method over the existing "direct method" of directly testing the interaction in a joint model.

Response: We appreciate the reviewer's positive comments.

I only have a few MINOR comments or clarification questions.

1. "a pseudo exposure" in the text and Figure 1 but also referred to as "a polygenic score" e.g. in Line 358. Should it be the same outcome Y but based on a GWIS? If so, perhaps better to refer it so and consistently in both the text and Fig 1; otherwise, how is the polygenic score constructed? And does it matter which PRS is used?

Response: We apologize for the confusion. A pseudo exposure refers to a polygenic score calculated based on the marginal effects of outcome Y. We agree with the reviewer's suggestion and have make our presentation consistent in both the text and Figure. 1. A notable advantage of our method is that constructing of the polygenic score is not needed..

We added a sentence "We do not need to construct this pseudo exposure in our analysis because we work directly on the summary statistics under the MR framework" in page 5 (line 12-13). In the Fig 1 legend, we added "...which can be viewed as a polygenic score for trait Y based on marginal effect sizes. However, our analysis does not require estimating this pseudo exposure."

2. What is/are the nominal significance level/levels being used in MR_GxE and each of the two tests/steps in the two-step procedure? Does it matter whether/how to adjust for multiple testing?

Response. In this approach, we first perform the T_{MR_GxE} , for which we set the nominal significance level at $5E-8$, a significance level used in GWAS. We next perform the direct test, and set the threshold for the direct test at $0.05/x$, where x is the number of independent variants significant in the T_{MR_GxE} test.

We added the following sentence "We set the significance level at 5×10^{-8} for the first step (T_{MR_GxE} test), and the significance level at $0.05/X$ for the step 2 T_{direct} test, where X is the number of independent significant variants in the first step/test." (page 5, line 27-29)

3. In the Discussion, it is stated that “ $T_{\{MR_GxE\}}$ is not affected by confounders such as population structure” and “It is less likely to have a confounding effect between a trait and its pseudo exposure”: Wouldn’t population structure (PS) influence most of MR? In particular, why wouldn’t PS influence IMRP?

Response. We appreciate the reviewer’s important question. Our goal is to detect $G \times E$ rather than to estimate the causal effect. Although the causal effect estimate can be biased if population structure is not well corrected, testing $G \times E$ by T_{MR_GxE} is not. The reason is that T_{MR_GxE} can be viewed as a weighted linear regression of the effect size of GWAS on the effect size of GWIS followed by searching for the outlying variants departing from the regression line. While the regression line (equivalent to causal effect estimate in MR) can be affected by population structure, detection of the outliers is not. Our method does not need to construct a polygenic score (also see the response to comment 1). This is in general true for either IMRP or other MR methods.

We have added the following paragraphs. “However, our goal is to detect $G \times E$ rather than to estimate the causal effect. Detecting $G \times E$ based on MR is less likely to be biased for these reasons:” (page 9, line 30-31)

“4) Although the causal effect estimate can be affected if population structure is not well corrected, testing $G \times E$ by T_{MR_GxE} is not. The reason is that T_{MR_GxE} can be viewed as a weighted linear regression of the effect size of GWAS on the effect size of GWIS followed by searching the outliers of variants departing from the regression line. While the regression line (equivalent to causal effect estimate in MR) can be affected by population structure, the outlier detection is not.”. (Page 9, line 36-41).

4. Is there any simple or intuitive explanation on why the proposed method is more powerful than the direct testing method? This is relevant because causal inference through MR seems to be a more difficult problem. Some comments would be helpful.

Response. We appreciate the reviewer’s important suggestion and comment. The proposed two-step method can be more powerful than the direct test for the following reasons. 1) the step 1 T_{MR_GxE} can increase power when a genetic variant has a mediation effect through the environmental factor. In this case, we expect a larger difference between the marginal effect and the main effect (See equation (3) in the main text) than no mediation. 2) The difference between the marginal and main effect can further increase when the environmental distributions between GWAS and GWIS cohorts are different (Equation (3) and Figure 1D). 3) In the two-step procedure, multiple comparison is significantly alleviated because only significant variants at step 1 need to be examined. (Page 8, line 18-28)

When we consider the PRS as the pseudo exposure, the causality of PRS on an outcome is already established. However, it is not our focus to estimate the causal effect of the PRS on the outcome. Our goal is to identify the horizontal pleiotropic variants, or outliers through IMRP software or other MR software, which can also be interpreted as $G \times E$. As we mentioned in

comment 3 above, the causal inference through MR may be biased, but the detection of pleiotropic variants is not (page 9, line 36-41).

Reviewer #2 (Remarks to the Author):

Zhu and colleagues present a new method for detecting GxE interactions (they do not appear to have a name for this new method, so I will simply refer to it as NewTest). The authors show that NewTest is well-calibrated (controls Type 1 error), and that it has the potential to be more powerful than the conventional test for GxE interactions (called GWIS). Specifically, NewTest has two potential advantages over GWIS. Firstly, NewTest uses a two-step approach, that first finds loci with significant non-additive effect, then considers only these loci when considering interaction with the environment (which reduces the multiple testing burden). Secondly, NewTest can combine GWIS results with those from marginal analysis (my instinct is that this enables more accurate estimation of the marginal effect, that in turn increases power of the interaction test).

Response: We appreciate the reviewer's positive comments. We are glad that the reviewer recognizes the advantages of our new two-step test (NewTest). Additionally, our new test can improve statistical power when the distribution of environmental factors is different in GWAS and GWIS study samples, and the testing genetic variants are also affected by the environmental factors under consideration (See our response to reviewer 1' comment 4.

It seems that (at present) the utility of NewTest is relatively slight. Specifically, the results presented indicate that, for the three lipid traits considered, the benefit of NewTest over GWIS is relatively small, and perhaps even negative (i.e., NewTest finds 10 significant loci, whereas GWIS found 15 or 17). Furthermore, to use NewTest, it is necessary to first perform a GWIS (so NewTest is not a standalone replacement for GWIS).

Response. We apologize for the confusion. Our NewTest is designed to test GxE interaction only, rather than the association between a genetic variant and the lipid traits. The 15 of 17 loci are only significant when including the main effects (by using a 2df joint test of the main effect and interaction). For these, GWIS failed to detect the interaction effect. For a fair comparison, we compared the NewTest (two-step procedure) with the direct test in GWIS. Among the 17 independent signals detected by T_{MR_GxE} , only two (BUD13 and LPL) were genome-wide significant by the direct GxE test. In comparison, the two-stage method identified 8 significant signals. (see our response to comment 4 by reviewer 1 and Table S3b)

Despite these criticisms, I am generally supportive of the paper. This is because I think the method is novel, and I am very interested by the connection between GWIS and MR.

Response. We appreciate the reviewer's support.

As an aside, while I consider the paper well-written, I nonetheless found it complicated, and so apologise if I have made a major error in my understanding of your work.

Response. We appreciate the reviewer's positive comment.

###

Major Comments

1 - I would appreciate a more explicit comparison of the powers of NewTest and GWIS on real data. For example, you could have a table reporting the number of SNPs significant from NewTest and GWIS (and the overlap) for each of the three traits.

*Response. The reviewer's suggestion is appreciated. We have added a new **Table S3b** reporting the number of significant SNPs for the two-step test and GWIS for the three traits and 4 environments. With this table, we can clearly see the advantage of the new test, which is able to detect more GxE signals.*

Table S3b. Comparisons of number of genome wide significant variants detected the two-step method and the direct method in multiancestry analysis (only variants with MAF>1% were counted).

	LDL x Curr Smk	LDL x Ever Smk	LDL x Curr Drink	LDL x Reg Drink	HDL x Curr Smk	HDL x Ever Smk	HDL x Curr Drink	HDL x Reg Drink	TG x Curr Smk	TG x Ever Smk	TG x Curr Drink	TG x Reg Drink
two-step	1	0	2	3	1	2	3	2	4	2	2	2
direct test	0	0	0	0	0	0	0	0	2	0	0	0

We have added “**In comparison to the direct test in GWIS, the two-step procedure identified more $G \times E$ signals for each of the three lipid traits and four environmental factors (Table S3b). This provides additional support for the enhanced statistical power of the two-step procedure.**” (Page 7, line 6-9)

2 - I explain above what I consider the two advantages of NewTest over GWIS. If my understanding is correct, can you quantify the relative importance of each. My opinion is that the first (reduction of multiple testing) is much larger than the second (ability to incorporate marginal results from larger GWAS). If so, this might be "good news", in the sense that I imagine there are relative few phenotypes where we have distinct results from GWIS and GWAS. For the real data results, it might be interesting to run NewTest using only the GWIS results (to see whether including GWAS results had much impact).

Response. The reviewer is correct that the reduction of multiple testing has larger impact than incorporating marginal results. We observed that among the 8 significant

signals detected by the two-stage test, only two signals (BUD13 and LPL for TG-current smoking) could be detected using the direct test in GWIS.

All the summary statistics for the marginal effects from GWIS samples were calculated by adjusting for the environmental factors (either smoking or alcohol drinking). In this case, the T_{MR-GxE} is theoretically identical to the direct test, therefore, the two stage-test reduces to the direct test in GWIS. This can also be observed in the real data analysis presented in Fig S1, where the correlation between the statistics of the T_{MR-GxE} and the direct test is 0.98 for LDL and current smoking. We thus did not run the two-stage test using GWIS data only. (Page 4, line 29-30)

3 - It seems to me, that for the real phenotypes considered, NewTest is less powerful than GWIS. If correct, this creates a contradiction with your simulations, where NewTest is almost always more powerful than GWIS (to use your notation, "Two-Step test" is more powerful than "Direct test"). This suggests your simulations are unrealistic (at least wrt to lipid traits). Given Fig 2D, it might suggest that the environmental factor tends to be very small for real traits.

Response. We apologize that we did not make our presentation clearer. From the real data analysis, we actually observed more significant signals using the two-step test than the direct test, which is consistent with our simulations (See Table S3b for real data comparison). As the reviewer noticed that the two-step procedure is more powerful when the environmental factor has higher mean in the GWAS than in the GWIS, or where there is mediation effect through the environmental factor or substantial reduction of multiple comparison burden. (see our response to comment 4 by reviewer 1 and Table S3b)

4 - I would appreciate if you made clear what you think the use of NewTest is. In my opinion, it is hard to view NewTest as a replacement for GWIS (because it gives, at most, a small increase in power, and requires a GWIS to be run first). If you agree, then I think good to make this explicit.

Response. We agree with the review that our two-stage test is not as a replacement for GWIS. We added "Considering the advantages of the two-step procedure, we view it as a complement rather than a replacement of the direct test. This perspective arises from the fact that the two-step test necessitates additional GWAS summary statistics and may be less powerful than the direct test in some situations (Fig. 1D)." (Page 8, line 31-34)

5 - Regarding the heritability estimates. Firstly, I am not aware of a paper that explicitly shows that it is valid to apply LDSC to GWIS summary statistics (and that the resulting estimate is an estimate of GxE heritability). The basic use of LDSC is to estimate h^2_{SNP} , and works because if SNP j contributes one unit of heritability, then SNP k will tag $cor(X_j, X_k)^2$ of this. Does the same

principal hold when considering interactions? (i.e., if SNP j contributes one unit of GxE heritability, then SNP k will tag $\text{cor}(X_j, X_k)^2$ of this?). Secondly, if it is valid to use LDSC to estimate GxE heritability, is there an advantage of using summary statistics from NewTest instead of those from GWIS (I can believe there is - perhaps using the former is better, because NewTest is less restrictive - but this is not clear from the text). If you retain the heritability analyses, I think it would be good to compare estimates using NewTest and GWIS summary statistics.

Response. We appreciate the reviewer's excellent comment, and we also agree with the reviewer that in general LDSC cannot be directly applied to GWIS summary statistics for estimating the GxE heritability. We did not estimate GxE heritability. Instead, we estimated the heritability using the summary statistics of T_{MR_GxE} , which is consistent with what the reviewer suggested. The estimated heritability is then interpreted as the lower bound of GxE heritability. It is currently unclear how to estimate GxE interaction heritability using GWIS summary statistics; thus, we did not perform this estimation. We have added the following: "Since the LDSC regression cannot be used to estimate $G \times E$ interaction heritability, our estimates reflect the low bound of the interaction and environmentally mediated heritability" (Page 9, line 4-6)

6 - Also regarding the heritability analysis. On Line 539, you say " We observed that some of the chromosome specific heritability estimates were negative. We only summed the non-negative chromosome specific heritability estimates." this is not satisfactory. Firstly, it sounds like you performed 22 separate analyses (one for each chromosome), then combined these. If so, it is not clear why (you say "to account for potential heterogeneity"). While there is a potential advantage in dividing by chromosome (e.g., if one chromosome has a much larger effect than others), I believe this should be done within a joint analysis (ie run LDSC once using 22 partitions) and not as 22 separate analyses. Further, it is generally bad practice to ignore negative estimates, as this leads to upward biases (instead, if you perform a joint analysis, you will get the per-chromosome estimates plus their sum, and a corresponding SD).

Response: We appreciate the reviewer's comments and suggestion. We have re-estimated the heritability in a joint analysis of all chromosomes as the reviewer suggested. The heritability estimate included the contributions from all chromosomes. We now report the new heritability estimates, which are substantially less than the previous estimated values, and we have updated Figure 4 accordingly. The heritability from GxE and mediation for HDL is no longer significant. We still observed significant heritability for LDL-C and ever smoking, as well as TG and smoking/alcohol drinking (Figure 4). Our abstract, main text and online methods correspondingly have been edited to reflect these changes.

Minor Comments

7 - L 102 - "Instead, the joint evidence of main genetic and GxE effects, in addition to the GxE

alone," - Do you mean, "by contrast" (you have just explained that few studies consider interactions, or at least not as the primary goal ... I think you are then saying GLI is one of the exceptions, in that it does focus on interaction)

Response: We have changed the text to "by contrast," as recommended (Page 3, line 18).

8 - L 115 - "Based on this principle, one can identify novel GxE using existing summary statistics without needing costly and time-consuming new analyses from all cohorts." I feel this downplays the resources needed - it remains that NewTest requires summary statistics from a GWIS (which as the authors have just explained, are not widely available).

Response: We made the following change according to the reviewer's suggestion. (Page 3, line 31)

"Based on this principle, one can identify novel $G \times E$ using existing available GWAS and GWIS summary statistics." (Page 3, line 31)

9 - L152 - not sure you have defined rho

Response: ρ is the mediation contribution of G through E (Page 4, line 18)

10 - L178 - "this pattern will not be impacted by the systemic variation across studies". It seems certain that experimental differences / noise, will have an impact, but I think your point is that it is unlikely to introduce systematic bias. So maybe this sentence should be "... we do not expect this pattern to be systematically impacted ..."

Response: We have made this change. (Page 5, line 7-8)

11 - L 194 - "applying ... to screening " - maybe "better" english is "by using ... to screen "

Response: We have made this change. (page 5, line 25-26)

12 - L 319 - "Our study demonstrated that the current heritability estimates based on marginal effects also include significant contributions from \times and mediation through the corresponding environment factors (Fig. 4 and Table S4)". It seems that you first discuss the heritability results in the Discussion (whereas I feel they should be in the results)

Response: As the reviewer observes, these findings should be described in the results section. We have now added the following text to the section $G \times E$ Interaction and Mediation to SNP Heritability". (page 7, line 41-44) "We observed significant interaction and mediation heritability ($P < 0.03$) with ever cigarette smoking for LDL-C, and alcohol consumption or cigarette smoking for TG, suggesting that the heritability estimates based on marginal effects also include significant contributions from $G \times E$ and mediation through the corresponding environment factors (Fig. 4 and Table S4)."

13 - L 443 - "No medication was present" (should be "mediation")

Response: We have made this change. (Page 12, line 18)

14 - L 495 - Please describe the charge summary statistics (i.e., that there are six sets, corresponding to GWIS of tg, hdl and ldl versus smoking or alcohol)

Response: We have made this change. (Page 19, line 13-16)

15 - L 500 - "we merged the" - perhaps better to say "aligned" (because else it sounds like you did a meta analysis of the GWAS and GWIS results)

Response: We have made this change. (Page 19, line 18)

16 - I really appreciated how much effort you put into the supplementary material.

Response: We appreciate the reviewer's comment.

REVIEWERS' COMMENTS

Reviewer #1 (Remarks to the Author):

The authors have carefully and satisfactorily address all my comments! Thanks and congrats!

Reviewer #2 (Remarks to the Author):

I am happy with the authors responses. In particular, thank you for adding the table in response to Comment 1, and for clearing up my confusion regarding Comment 3. Together, these indicate that the value of "NewTest" is a bit higher than I originally appreciated . Thank you also for taking into account my comments on the heritability analyses.

REVIEWERS' COMMENTS

Reviewer #1 (Remarks to the Author):

The authors have carefully and satisfactorily address all my comments! Thanks and congrats!

We appreciate the reviewer's positive comment.

Reviewer #2 (Remarks to the Author):

I am happy with the authors responses. In particular, thank you for adding the table in response to Comment 1, and for clearing up my confusion regarding Comment 3. Together, these indicate that the value of "NewTest" is a bit higher than I originally appreciated . Thank you also for taking into account my comments on the heritability analyses.

We appreciate the reviewer's positive comments.